# Temperature Gradient Method for Alleviating Bonding-Induced Warpage in a High-Precision Capacitive MEMS Accelerometer

**DOI:** 10.3390/s20041186

**Published:** 2020-02-21

**Authors:** Dandan Liu, Huafeng Liu, Jinquan Liu, Fangjing Hu, Ji Fan, Wenjie Wu, Liangcheng Tu

**Affiliations:** 1MOE Key Laboratory of Fundamental Physical Quantities Measurement, School of Physic, Huazhong University of Science and Technology, Wuhan 430074, China; liudandan@hust.edu.cn (D.L.); huafengliu@hust.edu.cn (H.L.); jinquanliu@hust.edu.cn (J.L.); fangjing_hu@hust.edu.cn (F.H.); fanji@hust.edu.cn (J.F.); 2Institute of Geophysics and PGMF, Huazhong University of Science and Technology, Wuhan 430074, China

**Keywords:** MEMS accelerometer, capacitive transducer, temperature gradient, bonding warpage

## Abstract

Capacitive MEMS accelerometers with area-variable periodic-electrode displacement transducers found wide applications in disaster monitoring, resource exploration and inertial navigation. The bonding-induced warpage, due to the difference in the coefficients of thermal expansion of the bonded slices, has a negative influence on the precise control of the interelectrode spacing that is essential to the sensitivity of accelerometers. In this work, we propose the theory, simulation and experiment of a method that can alleviate both the stress and the warpage by applying different bonding temperature on the bonded slices. A quasi-zero warpage is achieved experimentally, proving the feasibility of the method. As a benefit of the flat surface, the spacing of the capacitive displacement transducer can be precisely controlled, improving the self-noise of the accelerometer to 6 ng/√Hz @0.07 Hz, which is about two times lower than that of the accelerometer using a uniform-temperature bonding process.

## 1. Introduction

Recently, high-precision MEMS accelerometers are emerging for applications in precision gravity measurements [1], seismic monitoring [2], resource exploration [3] and inertial navigations [4], showing advantages in terms of both the size and the cost [5]. Specially, MEMS accelerometers using an area-variable periodic-electrode displacement transducer (APDT) are some of the most widely used MEMS accelerometers, due to their good linearity, high sensitivity and mature back-end electronics [5,6,7,8].

The APDT is formed by two electrode arrays on a moveless glass top cap and a matching electrode array on the movable silicon proof mass through a high-temperature bonding process [9]. Caused by the difference in the coefficients of thermal expansion (CTE) of glass and silicon, the bonding-induced stress and warpage are unavoidable. The uneven surfaces go against the precise control of the interelectrode spacing of the APDT, which is essential to achieve a high sensitivity for the displacement transducer, as demonstrated in our previous works [10,11]. In addition, the bonding-induced stress may damage the device or affect the long-term stability due to slow-releasing of the stress [12,13,14].

Various approaches have been reported for reducing the bonding-induced stress and warpage. The anchor was isolated from the spring-mass system by using a stress-isolation guard-ring structure in Reference [15]. However, it not only made the structure complicated, but also enlarged the size of the devices. A circular disk located in the center of the device and eight L-shaped elastic beams were used to isolate the bonding-induced stress in Reference [16]. However, it affect both the bonding strength and the structural stability. The bonding-induced warpage was reduced by selecting materials with similar CTE in References [17,18,19]. However, the method is not universal due to its limited choice of materials.

In this paper, we propose the theory, simulation and experimental demonstration of a new bonding strategy to alleviate the bonding-induced stress and warpage in a high-precision MEMS accelerometer. By applying different bonding temperatures to the bonded slices to adjust their shrinkage, the bonding induced warpage is reduced to quasi-zero. Being low-cost and convenient, the proposed bonding technology shows great prospect in industrial applications.

## 2. Design

A typical MEMS accelerometer using an APDT consists of a glass top cap and a silicon spring-mass system. Two columns of arrayed electrodes on the top cap and one column of arrayed electrodes on the spring-mass system form the APDT via a bonding process at a temperature over 504 K, as shown in Figure 1. A low-frequency acceleration variation (Δ*a*) is translated to the displacement of the proof-mass (Δ*x*) by
(1)$$\Delta x=-\frac{1}{{\left(2\pi f_{0}\right)}^{2}}\times \Delta a,$$
where *f*_0_ is the resonant frequency of the spring-mass structure. Then the displacement is theoretically sensed by the ADPT, given by
(2)$$\Delta C=\frac{2N\times \varepsilon \times l^{\prime }}{d}\times \Delta x$$
where Δ*C* is the capacitance variation of the APDT, *N* is the number of electrodes in one array, *l′* is the length of the electrodes and *d* is the spacing between the opposite electrode arrays [10]. It should be noted in Equation (2) that reducing the interelectrode spacing is essential for increasing the scale factor of the displacement transducer (*G_x−c_*). The capacitive changes can be detected using a lock-in amplifier circuit or highly sensitive low-noise quartz methods [20,21,22]. As the noise of the circuit (*Noise_electronics_*) is dominant in the self-noise of most MEMS accelerometers [11,23], increasing the scale factor of the capacitive transducer will lower the self-noise of the accelerometer (*Noise_accelerometer_*), which is given by
(3)$$Noise_{acceleration}=\frac{Noise_{electronics}}{G_{a-x}G_{x-c}}$$
where *G_a−x_* is the scale factor of the spring-mass system [11]. According to Equation (3), the noise of the accelerometer can be effectively decreased by reducing the interelectrode spacing. Since the warpage caused by the bonding process directly affects the interelectrode spacing, reducing the bonding warpage can effectively reduce the noise of the accelerometer.

When the accelerometer was cooled down to room temperature after the bonding process, the shrinkage of the top cap and the spring-mass system is respectively represented by
(4)$$S_{Top}=\bar{CTE_{glass}}\times \left(T-23\right),$$
and
(5)$$S_{spring}=\bar{CTE_{silicon}}\times \left(T-23\right),$$
where $\bar{CTE_{glass}}$ is the CTE of glass, *T* is the melting point of the solder and $\bar{CTE_{Silicon}}$ is the CTE of Silicon. Because of the difference in CTE for glass and silicon, the deformation of the top cap is different from the spring-mass system, which induces stress and surface warpage, as shown in Figure 2a. It is clearly illustrated that the bonding-induced warpage limits the precise control of the interelectrode spacing, which is of great importance to the sensitivity of the ADPT [10].

In order to alleviate the bonding-induced warpage, we propose a new bonding strategy by applying different high temperatures to the top cap and spring-mass system. The temperature difference (Δ*T*) is set based on the CTE difference of glass and silicon to provide the same shrinkage between the two slices, given by:(6)$$\bar{CTE_{glass}}\times \left(T-\Delta T-23\right)=\bar{CTE_{silicon}}\times \left(T-23\right).$$

According to Equation (6), the bonding-induced warpage and stress can be eliminated theoretically, as shown in Figure 2b.

## 3. Experiment and Simulation

Five samples with various temperature differences applied to the top cap and the spring-mass structure were bonded to validate the proposed method. The process flow started from a four-inch n-type silicon wafer with a thickness of 500 μm. A 200-nm-thick SiO2 layer was deposited on the surface (Figure 3). Firstly, the wafer was successively cleaned with acetone and isopropanol. Then, a patterned under-bump metal (UBM) consistings of a 40-nm-thick titanium (Ti) and a 200-nm-thick gold (Au) was deposited with electron beam evaporation followed by a lift-off process. Afterwards, a 100-nm-thick aluminium (Al) seed layer was deposited as conducting layer for electroplating [24]. Afterwards, a patterned Tin layer with a thickness of 20 μm was electroplated, followed by removing the Photoresist and seed layer by wet etching. The four-inch glass wafer (Borofloat 33) was fabricated by a similar process, except that the electroplated metal was gold with a thickness of 2.5 μm. The processed glass and silicon wafer were sawed into slices by a precision dicing machine (DS620). Finally, the glass slice and the silicon slices were bonded together by reflow soldering.

The bonding process was carried out in a FINEPLACER @ sigma which can directly show the real-time temperature of the silicon slice and glass slice. The temperature applied to the silicon slice was 504 K, which is the melting point of Tin. The temperature applied to the glass slice was different for the five samples, as shown in Table 1.

In order to verify the proposed bonding strategy, the bonding process was simulated by finite element analysis (FEA) using COMSOL Multiphysics. The three-dimensional geometry and the constraints were identical to the experimental model. The simulation consisted of two steps: Firstly, a thermal analysis was simulated by applying different temperatures on the surface of the glass and silicon based on the experiment. Secondly, the results of the thermal analysis were used as inputs for statics analysis, which obtains the bonding warpage. It should be noted that only elastic deformation was considered in the calculation and simulation for simplification.

## 4. Results

A fabricated sample was shown in Figure 4. The warpage of the samples was measured using a white light scanning profiler (Zygo NewViewTM7100). The surface topography of the samples was highly relevant to the temperature difference of the two bonded slices, as shown in Figure 5. The topography of the top caps had a hump-up tendency when the temperature applied to the top cap increased. The quantitative warpage was shown in Figure 6, which was defined by the height in the center subtracting that on the corner of the top cap’s surface. In order to verify the testing results, calculation results based on Equation (A3) of the Appendix A and the FEA results were used for comparison. The experiment, calculation and simulation results showed the same tendency that the warpage (positive when convex to the silicon and negative when convex to the glass) decreaseds with the enlargement of the glass temperature when the silicon temperature remains unchanged. When the temperature difference was −22 K, the warpage approached zero. The errors between the experimental, calculation and simulation results were within 23%. The experimental warpage was less than that of the calculation and simulation when the bonding warpage was large. The reason was probably the plastic deformation, which was not considered in the calculation and simulation, which would have released partial bonding stress. 

In order to verify the bonding strength of the proposed bonding process, all five samples were tested using a multi-function push-pull tester (XYZTEC @Condor Sigma Lite). Figure 7 shows the measured slice shear strength of the samples. Based on the MIL-STD-883E standards, the bonding strength was required to be greater than 5.0 kgf as the bonding area was 0.07 square inches. Hence the strength of all the samples using the proposed bonding process met the MLD-STD-883E standards.

Figure 8 shows the bonded chip and the assembled accelerometer with the circuit. The low-warpage bonding process was applied to the development of a high-precise MEMS accelerometer using an APDT that has been discussed in our previous publications [11]. The design and fabrication were the same as Reference [11], except for the bonding process. Beneficial from the flat surface by applying a proper temperature difference to the glass cap and the spring-mass structure, the interelectrode spacing was able to be controlled precisely. The accelerometer using the proposed bonding process had an interelectrode spacing of 8 μm, which was approximately two-thirds of that in accelerometer bonded with a uniform temperature. In order to calibrate the self-noise, the accelerometer bonded by the proposed strategy and another accelerometer bonded by a traditional process were tested statically in our cave laboratory with a quiet environment. A commercial seismometer (CMG-3EPS, GURALP) was installed adjacent to our accelerometers as a reference, as shown in Figure 9a. The results shows that the MEMS accelerometer bonded by the proposed strategy obtained an ultra-low self-noise of 6 ng/√Hz @ 0.07 Hz, which was two times lower than that of the accelerometer bonded using a uniform temperature, as shown in Figure 9b. The MEMS accelerometer is one of the most sensitive MEMS accelerometer in the world, as shown in Table 2. It is worth emphasizing that there is no need for addition mechanical structures, metal films or process steps for the proposed bonding strategy, which shows great prospect in industrial applications compared with those methods introduced in Section 1.

## 5. Conclusions

A new bonding strategy for alleviating the bonding-induced warpage was proposed in this paper, which effectively optimized a nano-g MEMS accelerometer. The effect of the bonding strategy was validated by calculation, simulation and experiment with errors within 23%. The interelectrode spacing of the proposed bonding process was 1.5 times better than the accelerometer bonded with a uniform temperature, and the noise floor was 6 ng/√Hz @ 0.07 Hz, which is two times better than other accelerometers. The strategy has no need for additional process steps and special materials, which allows us to use low-cost materials for bonding. In addition to accelerometer applications, the bonding technology also plays a key role in the semiconductor industry for forming multi-layer-structure devices and providing mechanical supports, as well as for ambient-disturbance isolation and input-output interfaces for MEMS devices or integrated circuits. The proposed strategy shows great prospect in industrial applications such as alleviating the bonding stress and warpage of MEMS-based sensors or integrated circuits as it is cheap, convenient and compatible with the MEMS process.

## Figures and Tables

**Figure 1 sensors-20-01186-f001:**
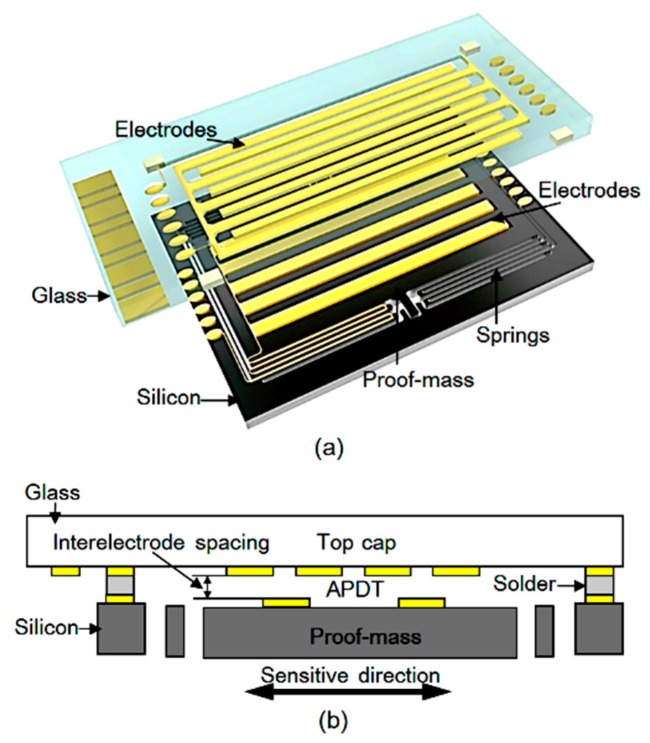
Schematic of the MEMS accelerometer using an area-variable periodic-electrode displacement transducer (APDT); (**a**) three-dimensional model; (**b**) cross-section view.

**Figure 2 sensors-20-01186-f002:**
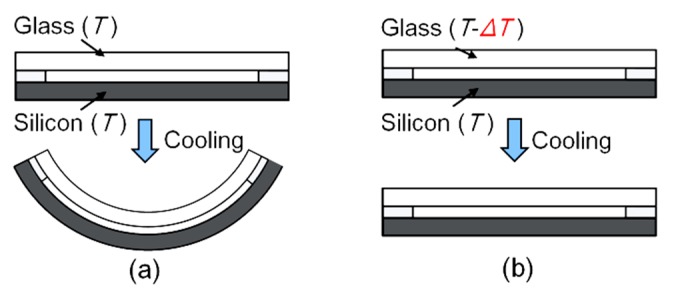
Schematic of bonding-induced warpage; (**a**) traditional bonding process with uniform high temperature; (**b**) the proposed bonding process with different high temperatures applied on the top cap and the spring-mass system.

**Figure 3 sensors-20-01186-f003:**
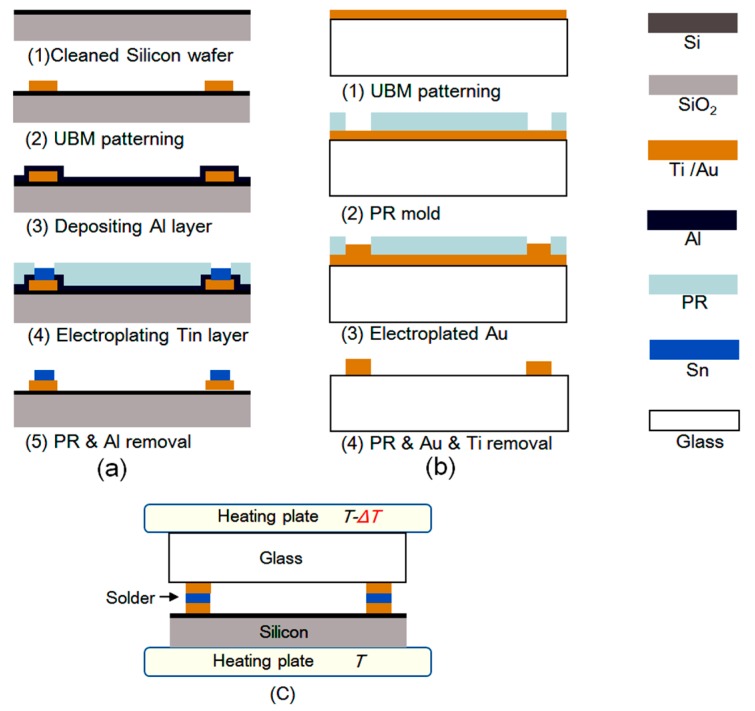
Process of the tested samples; (**a**) fabrication of the silicon slice; (**b**) fabrication of the glass slice; (**c**) bonding process with different temperature applied to the silicon slice and the glass slice.

**Figure 4 sensors-20-01186-f004:**
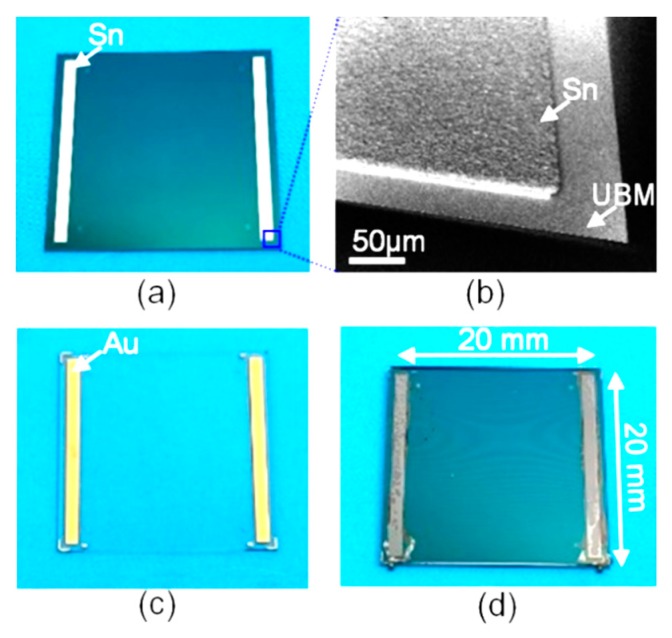
Photography of a fabricated sample; (**a**) silicon slice; (**b**) zoomed-in SEM picture of the Tin layer on the silicon slice; (**c**) glass slice; (**d**) bonded sample.

**Figure 5 sensors-20-01186-f005:**
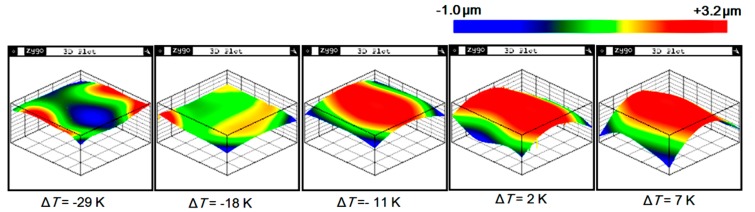
Topography of the glass slice in the tested samples with various temperature differences applied to the glass slice and the silicon slice.

**Figure 6 sensors-20-01186-f006:**
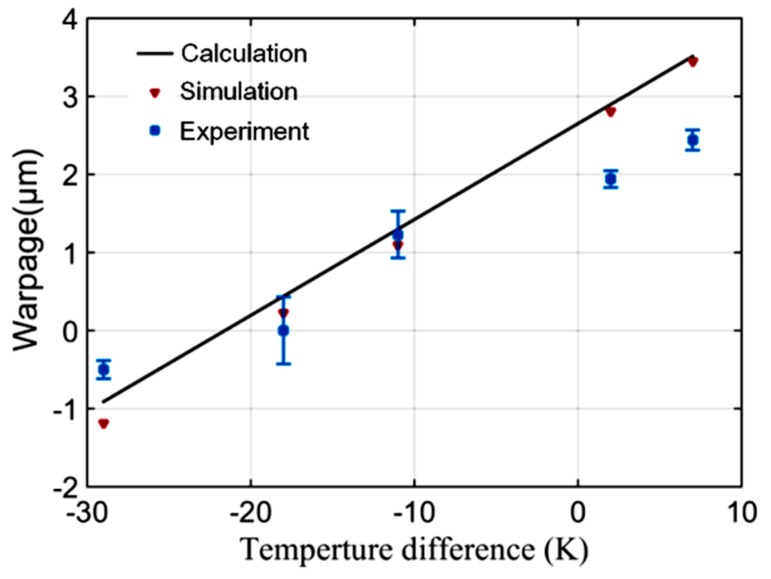
Experimental warpage of the tested samples with theoretical and simulated results as a reference.

**Figure 7 sensors-20-01186-f007:**
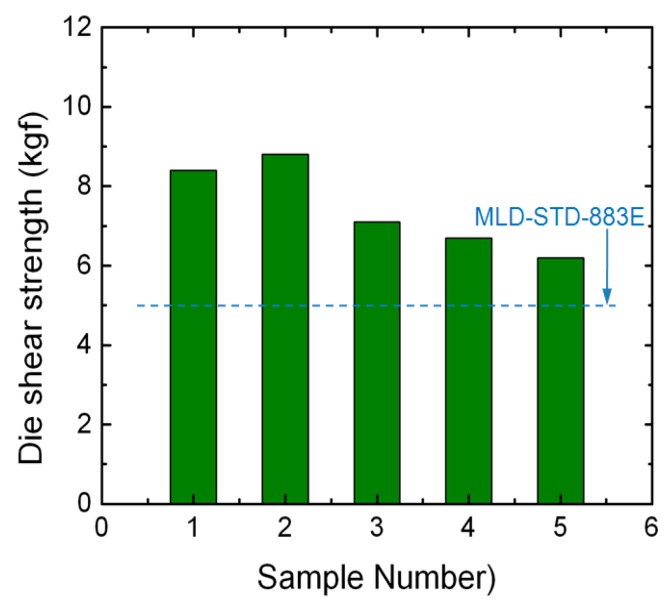
Shear strength of the samples bonded with different temperature gradients.

**Figure 8 sensors-20-01186-f008:**
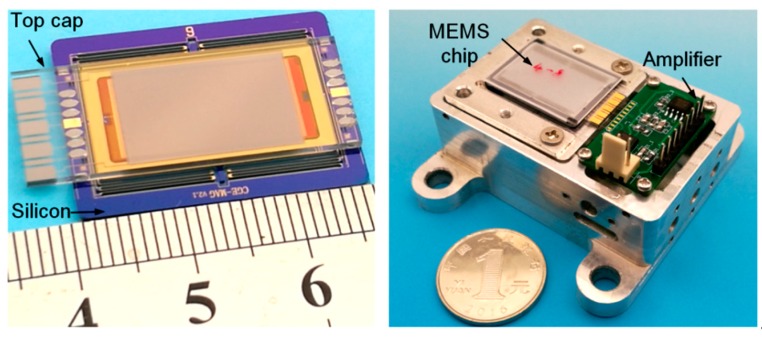
MEMS accelerometer using the proposed bonding process.

**Figure 9 sensors-20-01186-f009:**
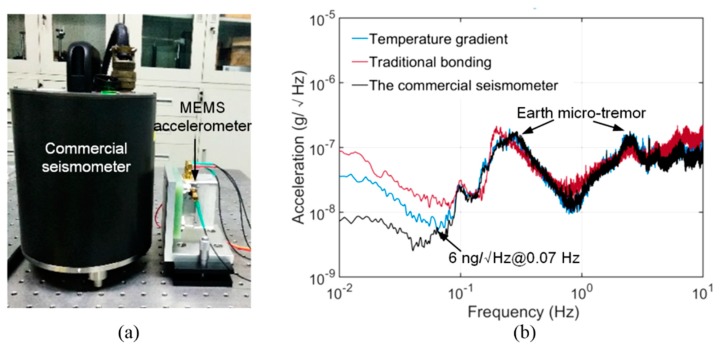
(**a**) Calibrating the self-noise of the MEMS accelerometer with a commercial seismometer for reference. (**b**) The power spectral density (PSD) of the output of the MEMS accelerometers and the commercial seismometer. The earth micro-tremor is dominant in the noise floor for the bandwidth above 0.1 Hz. For the bandwidth below 0.05 Hz, the self-noise of the sensors is dominant.

**Table 1 sensors-20-01186-t001:** Temperature applied to the silicon slice and glass slice.

No.	1	2	3	4	5
Si	504 K	504 K	504 K	504 K	504 K
Glass	475 K	486 K	493 K	506 K	511 K
Δ*T (T_Si_–T_glass_*)	−29 K	−18 K	−11 K	+2 K	+7 K

**Table 2 sensors-20-01186-t002:** The self-noise comparison between the proposed MEMS accelerometer and typical high-precision MEMS accelerometers.

Accelerometers	University of Glasgow [1]	Imperial College [23]	Hewlett Packard [3]	This Work
Applications	Gravimeter	Seismic Sensor	Seismic Sensor	Acceleration Sensor
Self-noise (ng/√Hz)	10	2	10	6

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
