# Peer review of "Temperature Gradient Method for Alleviating Bonding-Induced Warpage in a High-Precision Capacitive MEMS Accelerometer"

_sensors, 2020, doi:10.3390/s20041186_

Round 1

Reviewer 1 Report

The paper presents interesting results to reduce warpage by bonding with different temperatures. The experimental part is well presented and relevant.  

There are some aspects to be improved, clarified:

The FEA model mentioned in the paper is not presented. How the model was build (geometry, boundary conditions)? Is a simplified model or is identical with the experimental model?   The error between the results are different for different temperatures that you explain by incomplete consideration on the deformation of Tin layer in the model. If is an effect that is not temperature dependent why is affecting different the model for different temperatures? Extended comments on Figure 6 should be included. The conclusions doesn't include any comments related the results between the experiment and analytical and FEA results.   Based on your models there are the possibility to predic the warpage and to estimate the temperature difference necessary to be applied on bonding slices? Such a model will be very useful for industrial applications. 

Author Response

Thank you for your professional comments on our study. We agreed with your suggestions for improving the paper and modified them one by one.

1.The FEA model mentioned in the paper is not presented. How the model was build (geometry, boundary conditions)? Is a simplified model or is identical with the experimental model? 

Response: We agree that the FEA model should be described in details. The FEA model is identical with the experimental model.

Modifications: The FEA model is described in details from Line 122 to Line 128. The headline of section 3 is modified to ‘Experiment and Simulation’ correspondingly.

2.The error between the results are different for different temperatures that you explain byincomplete consideration on the deformation of Tin layer in the model. If is an effect that is not temperature dependent why is affecting different the model for different temperatures? Extended comments on Figure 6 should be included.

Response: The errors that are different for different temperatures are probably caused by the plastic deformation of the Tin layer, as only elastic deformation was considered in calculation and simulation for simplification. The plastic deformation of the Tin layer releases the bonding stress, which will make the experimental warpage less than that of calculation and simulation. Because the plastic deformation occurs when the stress is beyond the yield strength, the effect of the plastic deformation is more remarkable when the bonding warpage is large. Another reason that contributes to the errors is the uncertain physical properties of the Tin layer that is related to multi-conditions during the reflowing process.

Modifications: The simplification is clarified from Line 127 and Line 128. Extended comments with respect to the errors in Figure 6 are supplemented (Line 142 to Line 144).

3. The conclusions doesn't include any comments related the results between the experiment and analytical and FEA results. 

Response: Thank you for your guidance and We agree that the comments should be included.

Modifications: Comments related to the results between the experiment, calculation and FEA results are added to the Conclusion Section (Line 195 and Line 196).

4.Based on your models there are the possibility to predic the warpage and to estimate the temperature difference necessary to be applied on bonding slices? Such a model will be very useful for industrial applications. 

Response: Thank you for your professional comments on our study. It is possible to predict the warpage and to estimate the temperature difference necessary to be applied on bonding slices based on the proposed model, as verified in this paper. We believe the model is useful for industrial applications as it is effective and compatible to MEMS process without the need for addition mechanical structures, deposited films or process steps .

Reviewer 2 Report

Comments to the authors:

I suggest same corrections.

Page 1: Title: Authors should check spelling (uppercase and lowercase).

Page 2:  lines 68- 75:  The authors should include: “The detection of small capacitive changes can also be made using highly sensitive low noise quartz methods, as shown in the following articles:

-Matko, V.; Milanovic, M. Temperature-compensated capacitance-frequency converter with high resolution. Sensor Actuat a-Phys 2014, 220, 262-269.

-Ivanov, Vadim. Direct electro-optic effect in langasites and α-quartz. Optical Materials. May 2018 79:1-7.

-Matko V., Šafarič R. Major improvements of quartz crystal pulling sensitivity and linearity using series reactance. Sensors, 2009, 9, 10, 8263-8270.

Page 6: Results: Could authors get any comment on long-term stability due to slow-releasing of the strees in proposed method.

Author Response

Thank you for your professional comments on our study. We agreed with your suggestions for improving the paper and modified them one by one.

I suggest some corrections.

1. Page 1: Title: Authors should check spelling (uppercase and lowercase).

Response: Thank you for your correction.

Modifications: The title is corrected (Line 2 to Line 4).

2. Page 2:  lines 68- 75:  The authors should include: “The detection of small capacitive changes can also be made using highly sensitive low noise quartz methods, as shown in the following articles:

-Matko, V.; Milanovic, M. Temperature-compensated capacitance-frequency converter with high resolution. Sensor Actuat a-Phys 2014220, 262-269.

-Ivanov, Vadim. Direct electro-optic effect in langasites and α-quartzOptical Materials. May2018 79:1-7.

-Matko V., Šafarič R. Major improvements of quartz crystal pulling sensitivity and linearity using series reactance. Sensors2009, 9, 10, 8263-8270.

Response: We are sorry that some important methods were not included in the manuscript.

Modifications: These methods are discribed from Line 68 to Line 70 with the articles cited.

3. Page 6: Results: Could authors get any comment on long-term stability due to slow-releasing of the strees in proposed method.

Response: The results in this paper were not able to support the performance of long-term stability. Evaluating the effect of the slow-releasing of the stress is a great challenge, as the long-term stability is related to multi-factors including the environmental disturbs. 

The influence of the slow-releasing of the stress on long-term stability has been verified in many literatures. One way for measuring the stress of multi-layer structures is measuring the warpage, followed by calculation using the geometric dimensions and material properties. As the warpage was reduced by the proposed method, we believe that the proposes method is beneficial to the long-term stability.

Reviewer 3 Report

The study reported in this article is relevant and interesting. I have minor comments as follws

1. In title should be informed that is a capacitive MEMS accelerometer.

2. References to the equations presented in section 2 are missing.

3. There are some typos in the manuscript text. For example: "3. Experment"

4. Lack information on the used wafers. What is the diameter? n-type or p-type?

5. The characteristics of the glass wafer were not described.

6. The discussion of the results shown in section 4 is quite superficial. The results are only described. There is no discussion in deep. Also the results obtained were not compared to those described in the literature for capacitive MEMS accelerometer, especially in relation to other approaches used for reducing the bonding-induced stress and warpage.

7. The quality and resolution of the figures can be improved.

Author Response

Thank you for your professional comments on our study. We agreed with your suggestions for improving the paper and modified them one by one.

The study reported in this article is relevant and interesting. I have minor comments as follows

1. In title should be informed that is a capacitive MEMS accelerometer.

Response: We agree with the reviewer’s opinions.

Modifications: The title is modified as ‘Temperature Gradient Method for Alleviating Bonding-induced Warpage in a High-precision Capacitive MEMS Accelerometer’ (Line 2 to Line 4).

2. References to the equations presented in section 2 are missing.

Response: We agree with the reviewer’s opinions.

Modifications: The references to the equations presented in section 2 are added at Line 67, Line 74 and Line 97, respectively. In addition, the literatures of Equation (2) and Equation (3) are cited at Line 67 and Line 74, respectively.

3.There are some typos in the manuscript text. For example: "3. Experment"

Response: Thank you for your correction. We checked the whole paper carefully to find out the typos.

Modifications:

noted~ be noted (Line 67)

,~. (Equation (3))

,~. (Equation (6))

Experment~ Experiment (Line 99)

error~ errors (Line 141)

was~ were (Line 142)

4. Lack information on the used wafers. What is the diameter? n-type or p-type?

Response: We agree that the information of the silicon wafer should be discribed.

Modifications: The information of the silicon wafer is supplemented from Line 102 to Line 104.

5. The characteristics of the glass wafer were not described.

Response: We agree that the characteristics of the glass wafer should be described.

Modifications: The types of the glass wafer is described at Line 109.

6. The discussion of the results shown in section 4 is quite superficial. The results are only described. There is no discussion in deep. Also the results obtained were not compared to those described in the literature for capacitive MEMS accelerometer, especially in relation to other approaches used for reducing the bonding-induced stress and warpage.

Response: We agree with the reviewer’s opinions.

Modifications: A deep analysis of the results is supplemented from Line 142 to Line 145. The self-noise of the proposed accelerometer, which is a key parameters for describing the precision, is compared with typical high-precision MEMS accelerometers in Table 2 and the main text from Line 177 to Line 181. In addition, the main advantages over the other approaches for reducing the bonding-induced stress and warpage are concluded from Line 178 to Line 181.

7. The quality and resolution of the figures can be improved.

Response: We agree with the reviewer’s opinions .

Modifications: Figure 7 is redrawn for improving the quality. In addition, the brightness and contrast of all these figures are adjusted.

Round 2

Reviewer 1 Report

The paper was improved and all the comments have been addressed in the revised version